# Two Signaling Modes Are Better than One: Flux-Independent Signaling by Ionotropic Glutamate Receptors Is Coming of Age

**DOI:** 10.3390/biomedicines12040880

**Published:** 2024-04-16

**Authors:** Valentina Brunetti, Teresa Soda, Roberto Berra-Romani, Giovambattista De Sarro, Germano Guerra, Giorgia Scarpellino, Francesco Moccia

**Affiliations:** 1Laboratory of General Physiology, Department of Biology and Biotechnology “L. Spallanzani”, 27110 Pavia, Italy; valentina.brunetti01@universitadipavia.it (V.B.); giorgia.scarpellino@unipv.it (G.S.); 2Department of Health Sciences, School of Medicine and Surgery, Magna Graecia University of Catanzaro, 88100 Catanzaro, Italy; teresa.soda@unicz.it (T.S.); desarro@unicz.it (G.D.S.); 3Department of Biomedicine, School of Medicine, Benemérita Universidad Autónoma de Puebla, Puebla 72410, Mexico; rberra001@hotmail.com; 4System and Applied Pharmacology@University Magna Grecia, Science of Health Department, School of Medicine, Magna Graecia University of Catanzaro, 88110 Catanzaro, Italy; 5Department of Medicine and Health Science “Vincenzo Tiberio”, School of Medicine and Surgery, University of Molise, 86100 Campobasso, Italy; germano.guerra@unimol.it

**Keywords:** glutamate, ionotropic glutamate receptors, AMPA receptors, kainate receptors, NMDA receptors, flux-independent signaling, non-canonical signaling

## Abstract

Glutamate is the major excitatory neurotransmitter in the central nervous system. Glutamatergic transmission can be mediated by ionotropic glutamate receptors (iGluRs), which mediate rapid synaptic depolarization that can be associated with Ca^2+^ entry and activity-dependent change in the strength of synaptic transmission, as well as by metabotropic glutamate receptors (mGluRs), which mediate slower postsynaptic responses through the recruitment of second messenger systems. A wealth of evidence reported over the last three decades has shown that this dogmatic subdivision between iGluRs and mGluRs may not reflect the actual physiological signaling mode of the iGluRs, i.e., α-amino-3-hydroxy-5-methyl-4-isoxasolepropionic acid (AMPA) receptors (AMPAR), kainate receptors (KARs), and N-methyl-D-aspartate (NMDA) receptors (NMDARs). Herein, we review the evidence available supporting the notion that the canonical iGluRs can recruit flux-independent signaling pathways not only in neurons, but also in brain astrocytes and cerebrovascular endothelial cells. Understanding the signaling versatility of iGluRs can exert a profound impact on our understanding of glutamatergic synapses. Furthermore, it may shed light on novel neuroprotective strategies against brain disorders.

## 1. Introduction

The distinction between ionotropic and metabotropic receptors was introduced in 1979 by Eccles and McGeer [1] and is based on the distinct signaling modes of neurotransmitter receptors. Ionotropic receptors are ligand-gated channel proteins that directly gate the flow of ions across the plasma membrane (PM), thereby leading to rapid postsynaptic excitation or inhibition, typically lasting a few milliseconds. Metabotropic receptors induce the postsynaptic response via G-proteins, which modulate ion channel activity either directly or via intracellular second messengers. Metabotropic receptors are therefore also termed G-protein coupled receptors (GPCRs) and are responsible for slow synaptic transmission [2]. Metabotropic receptors also include tyrosine kinase receptors (TKRs) and guanylate cyclase receptors [3], but they do not play a primary role in synaptic transmission. Over the last three decades, however, it has been increasingly recognized that this traditional classification no longer reflects the versatile intracellular signaling pathways that can be mediated by ionotropic receptors [4,5,6,7,8]. Several neurotransmitters, including glutamate, acetylcholine, and γ-aminobutyric acid (GABA), bind to both ionotropic and metabotropic receptors in the central nervous system [8,9]. However, unexpected evidence has shown that the ionotropic receptors do not simply act as neurotransmitter-gated ion-conducting pores whose opening results in rapid membrane depolarization (e.g., glutamate or acetylcholine) or hyperpolarization (e.g., GABA). Ionotropic receptors can also respond to ligand binding by recruiting multiple downstream signaling pathways via the functional interaction with effector proteins that do not require ion flux [4,5,6,7,8,10,11,12]. Thus, ionotropic receptors can engage “non-canonical” signaling pathways that are activated independently of their “canonical” mode of action and regulate a different panel of cellular functions [8]. The non-canonical signaling mode of ionotropic receptors is also known as metabotropic-like, as it requires a conformational change in the receptor protein that directly activates an intracellular second messenger pathway [8]. At a recent online meeting sponsored by the American Society for Biochemistry and Molecular Biology (ASBMB), it was suggested that these definitions can be replaced with the term “flux-independent” signaling [7]. Herein, we aim at providing a comprehensive view of the flux-independent signaling mechanisms by which ionotropic glutamate receptors (iGluRs) regulate a broad range of functions not only in neurons but also in non-excitable cells, such as brain astrocytes and endothelial cells.

## 2. Flux-Independent Signaling by iGluRs

Glutamate is the primary excitatory neurotransmitter in the brain [13] and also serves as a neuromodulator to couple rhythmic activity with synaptic transmission in neuronal networks [14]. The postsynaptic response to glutamate is mediated by iGluR and metabotropic glutamate receptors (mGluRs). The iGluRs consist of tetrameric non-selective cation channels that can be further subdivided into three sub-families [13,15]: α-amino-3-hydroxy-5-methyl-4-isoxasolepropionic acid (AMPA) receptors (AMPARs), kainate receptors (KARs), and N-methyl-D-aspartate (NMDA) receptors (NMDARs). The iGluRs share a similar molecular structure, including the topology of the ion-conducting pathway and the ability to open rapidly upon glutamate association with an extracellular ligand binding domain [15]. However, whereas AMPARs (and possibly KARs) only conduct inward Na^+^ currents, NMDARs are also permeable to extracellular Ca^2+^ and can directly engage Ca^2+^-dependent signaling pathways [9,16]. The synaptic release of glutamate activates a fast excitatory postsynaptic current (EPSC) that is mediated by AMPARs (AMPAR-EPSC) and followed by a slower NMDAR-mediated EPSC (NMDAR-EPSC) [17]. The mGluRs are GPCRs that are also subdivided into three main groups based on their synthetic agonist sensitivity, G-protein coupling, and sequence homology: Group 1, consisting of mGluR1 and mGluR5, which mainly stimulate phospholipase Cβ (PLCβ) via G_q/11_ protein; Group 2, consisting of mGluR2 and mGluR3; and Group 3, consisting of mGluR4, mGluR6, mGluR7, and mGluR8, which inhibit adenylate cyclase (AC) via G_i/o_ proteins and modulate several ion channels via the release of the Gβγ dimer [13,18]. Based on their sequence similarity, a fourth sub-family of iGluRs was discovered in the mid-1990s, i.e., the δ (GluD) receptors, which exist in two isoforms, GluD1 and GluD2 [19]. GluD receptors do not bind glutamate and were long considered as “orphan” receptors until it was discovered that D-serine was able to activate them [20]. However, GluD receptors do not function as canonical ionotropic receptors but rather associate with trans-synaptic protein complexes that primarily trigger metabotropic-like signaling pathways in the postsynaptic terminals [19,21]. The ion conduction pathway of GluD receptors, however, is not blocked, as evidenced by the inward currents mediated by GluD1 and GluD2 following the stimulation of Group 1 mGluRs [22,23]. We refer the reader to several recent reviews that provide a comprehensive and exhaustive description of the structural and mechanistic features of GluD receptors [8,19,21]. Interestingly, parallel studies have shown that canonical iGluRs can also signal in a flux-independent manner in excitatory pyramidal neurons, inhibitory interneurons, astrocytes, and cerebrovascular endothelial cells [6,7,8,9,24]. Thus, the ability to utilize both flux-dependent and flux-independent signaling modes is an integrative feature of glutamate-gated iGluRs.

## 3. Flux-Independent Signaling by AMPARs

AMPARs are composed of the assembly of different combinations of GluA1-GluA4 subunits that are encoded by the *Gria1-4* genes [13,15]. Heteromeric AMPARs are the main iGluRs that mediate the fast, excitatory synaptic transmission in the brain and relieve NMDARs from Mg^2+^-dependent inhibition at glutamatergic synapses [13,15]. GluA2 and GluA3 are the most abundant AMPAR isoforms in glutamatergic pyramidal neurons, while inhibitory GABAergic interneurons mainly express GluA1 and a small amount of GluA2 and GluA3 [25,26]. It has long been thought that most AMPAR assemblies include GluA2/GluA3 and GluA1/GluA2 heterodimers [27,28]; however, a recent investigation demonstrated that native AMPARs may also be heterotrimeric channels composed of GluA1, GluA2, and GluA3 subunits [29]. The GluA2 messenger ribonucleic acid (mRNA) can be edited during fetal development: a single aminoacidic residue, i.e., glutamine (Q) (Ca^2+^-permeable), can be replaced by arginine (R) (Ca^2+^-impermeable) within the ion pore. Incorporation of the edited GluA2 into the heteromeric AMPAR structure abolishes its inward rectification and strongly reduces its Ca^2+^ permeability. Conversely, AMPARs that lack GluA2 mediate both Na^+^ and Ca^2+^ entry [16]. The role of Ca^2+^-permeable AMPARs in the regulation of receptor trafficking, synaptic plasticity, learning, and memory has been increasingly recognized [30]. Moreover, some auxiliary proteins, such as stargazin and cornichon-2, can increase the Ca^2+^ permeability of AMPARs by subtly modifying the narrow constriction of the filter pore [16]. In addition to the biophysical heterogeneity of their ionotropic signaling, AMPARs can also signal in a flux-independent manner [7,8].

### 3.1. Flux-Independent Signaling by AMPARs Involves Multiple Metabotropic Signaling Pathways

AMPARs were the first iGluR shown to signal in a flux-independent mode [31]. Early work showed that the selective agonist AMPA induced AMPARs to inhibit AC activity through the recruitment of G_i/o_ proteins either directly or via an indirect association with an adaptor protein (Figure 1) [31]. This seminal discovery paved the way for subsequent studies aimed at assessing the functional consequences of flux-independent signaling by AMPARs. AMPARs can recruit G_i/o_ proteins to activate the phosphoinositide-3-kinase–protein kinase B/Akt (PI3K-PKB/Akt) and the extracellular signal-regulated kinase (ERK) pathways, which induce rat cerebellar granule cell (GrC) survival (Figure 1) [32]. Additionally, in rat hippocampal neurons, AMPARs trigger the association of the G_i/o_ protein-derived β subunit with voltage-gated Na^+^ channels, thereby inducing Na^+^ entry that promotes mitochondrial Ca^2+^ release by reversing the mitochondrial Na^+^/Ca^2+^ exchanger (Figure 1) [33]. Furthermore, flux-independent signaling by AMPARs stimulates the migration of rat oligodendrocyte progenitor cells by inducing the formation of a α_V_/myelin proteolipid protein complex, which in turn elicits a pro-migratory oscillatory Ca^2+^ signal [34,35]. Recruitment of G_i/o_ proteins is also used by AMPARs to inhibit the expression of the *Arc* gene (Figure 1) [36], which is critical for maintaining synaptic potentiation and long-term consolidation of memory [37]. However, the primary role of flux-independent AMPAR signaling is to modulate synaptic activity [8]. Climbing fibers (CFs) project from the inferior olive to provide an excitatory input to the molecular layer of the cerebellum, where they induce complex spikes that are critical for motor behavior [38]. Presynaptic AMPARs are activated by CFs to restrain GABA release from cerebellar interneurons, i.e., stellate cells and basket cells, by inhibiting voltage-gated Ca^2+^ channels (Ca_V_2.1) through G_i/o_ proteins (Figure 1) [39]. The same flux-independent pathway underlies AMPAR-mediated inhibition of presynaptic Ca_V_2.1 currents and glutamate release from the calyx held in the rat auditory brainstem [40]. The flux-independent activity of AMPARs can also suppress the nitric oxide (NO)-induced, cyclic guanosine monophosphate-gated currents in rat retinal ganglion cells, as well as playing a role in visual processing [41].

AMPARs can also signal in a flux-independent mode by interacting with Lyn (Figure 1) [42], a protein tyrosine kinase of the Src family [8], in primary rat cerebellar cells. GluA2 physically associates with Lyn, which activates the ERK pathway and induces the expression of brain-derived neurotrophic factor (BDNF) (Figure 1) [42]. In addition, the AMPAR-mediated activation of ERK leads to synapsin I phosphorylation, which enhances synaptic vesicle recycling and likely contributes to the regulation of short-term plasticity (Figure 1) [43].

### 3.2. The Molecular Determinants of Flux-Independent Signaling by AMPARs

The molecular basis of AMPAR-induced recruitment of distinct flux-independent pathways is far from being understood. The modular architecture of AMPARs includes four domains, i.e., the extracellular NH_2_-terminal domain (NTD), the external ligand-binding domain, the transmembrane domain, and the cytosolic COOH-terminal domain (CTD). The CTD enables interaction with several auxiliary subunits that regulate AMPAR trafficking and gating, but little information is available about the components of the AMPAR interactome that are recruited to generate flux-independent signals [44,45]. Preliminary reports suggest that the AMPAR interactome does not include signaling proteins, with the exceptions of the G_i/o_ protein and Lyn [46,47]. Nevertheless, knowledge of flux-independent AMPAR signaling is likely to be expanded by unexpected findings derived from the investigation of other signaling cascades. For instance, store-operated Ca^2+^ entry (SOCE) is a ubiquitous voltage-independent Ca^2+^ entry pathway that is activated upon depletion of the endoplasmic reticulum (ER) Ca^2+^ store to refill the ER lumen with Ca^2+^. SOCE is mediated by the physical association between STIM and Orai, which, respectively, serve as sensors of the ER Ca^2+^ concentration and Ca^2+^ entry channel on the PM in the main cellular components of the neurovascular unit, i.e., neurons, astrocytes, and endothelial cells [48,49,50,51,52,53]. Surprisingly, STIM proteins can interact with both GluA1 and GluA2 in rat cortical neurons and astrocytes. In addition, the pharmacological blockade of AMPARs with 2,3-dihydroxy-6-nitro-7-sulphamoyl-benzo(F)quinoxaline and cyanquixaline (6-cyano-7-nitroquinoxaline-2,3-dione) strongly reduced SOCE in both cell types [54]. This study did not assess whether the AMPARs that support SOCE include the GluA2(Q) (Ca^2+^-permeable) or the GluA2 (R) (Ca^2+^-impermeable) subunit, but it provided clear-cut evidence that AMPARs interact with other signaling proteins and that this interaction does not require ligand binding, as is also highlighted in the next section.

### 3.3. Flux-Independent Signaling by AMPARs May Involve a Structural Component

The extracellular NTD of GluA2 can bind to the presynaptic and postsynaptic cell-adhesion protein N-cadherin to form a trans-synaptic scaffolding complex that promotes synaptogenic signaling and facilitates synaptic spine formation in rat primary hippocampal cortical neurons [55,56]. Accordingly, GluA2 overexpression increases spine density and width in primary cortical pyramidal neurons and also induces the appearance of dendritic spine-like formations in GABAergic interneurons, which normally lack this type of PM protrusion [56]. In agreement with these findings, GluA2 is also required to promote spine formation in the subset of hippocampal and cortical GABAergic neurons that carry dendritic spines [8,57,58]. By contrast, the overexpression of a GluA2 protein mutant lacking the NTD inhibits spine morphogenesis [56]. The interaction between GluA2 and N-cadherin is enabled by human natural killer (HNK)-1 [59,60], a glyco-epitope expressed by some cell adhesion molecules in the nervous system [61]. Thus, the genetic suppression of either HNK-1 [60] or GluA2 [62] leads to a reduction in the number of mushroom-like mature spines and an increase in the number of immature filopodium-like spines. In accord with this, the physical association between the NTD of GluA2 and N-cadherin sustains long-lasting synaptic plasticity in the hippocampus. Genetic deletion of GluA2, but not GluA3, prevents mGluRs-dependent long-term depression (LTD) at the Schaffer collateral (SChC)-CA1 synapse: the GluA2/N–cadherin interaction is required to activate the Rho GTPase Rac1 and thereby induce cofilin-dependent actin reorganization [63]. Furthermore, the loss of HNK-1 expression impairs NMDAR-dependent long-term potentiation (LTP) and spatial memory formation at the same synaptic contact in the hippocampus [64]. Intriguingly, only GluA2(Q)-containing AMPARs, which are expressed in the developing cortex, are able to stimulate spine morphogenesis [65]. Therefore, the GluA2(Q) variant may be critical for regulating developmental synaptogenesis by promoting trans-synaptic signaling that does not require agonist binding [8]. However, the NTD of postsynaptic GluA2-containing AMPARs can also stabilize the presynaptic terminal in a flux-independent manner in adult rat cortical [66] and hippocampal [67] pyramidal neurons. Nevertheless, the trans-synaptic signal conveyed to presynaptic terminals does not require the interaction with N-cadherin in mature synapses [67]. These findings suggest that the structural signaling by GluA2-containing AMPARs contributes to spine formation in a developmentally regulated manner and to synaptic stabilization in the adult in both the cortex and hippocampus. In addition, it has recently been shown that the upregulation of HNK-1 also increases GluA2 expression in the cerebellum [68]. Future work will have to assess whether GluA2 is also able to transmit trans-synaptic signals aimed at stabilizing synaptic contacts in this brain structure, which plays a crucial role not only in motor control and motor learning [69,70,71,72,73], but also in cognitive functions [38].

## 4. Flux-Independent Signaling by KARs

KARs are composed of the assembly of various combinations of GluK1-GluK5 subunits that are encoded by the *Grik1-5* genes. The low-affinity GluK1-GluK3 subunits can form homomeric or heteromeric receptors that are gated by high concentrations of glutamate and the selective agonist kainate (KA). The high-affinity GluK4 and GluK5 are sensitive to lower agonist concentrations and combine with GluK1-GluK3 to form di-, tri-, or tetra-heteromeric receptors [74,75,76,77]. The molecular heterogeneity of KARs is enhanced by the existence of multiple GluK1-GluK3 splice variants and by the pre-mRNA editing of GluK1 and GluK2 subunits at the Q/R site [8,77]. The Q/R editing of GluK2 pre-mRNA by the nuclear enzyme ADAR2 reduces single-channel conductance and Ca^2+^ permeability and decreases KAR surface expression [78,79]. In addition, either presynaptic or postsynaptic KARs can interact with an increasing number of proteins, including the two integral transmembrane proteins, Neuropilin Tolloid-like 1 and Neuropilin Tolloid-like 2 (Neto1 and Neto2) [77].

KARs are widely distributed throughout the nervous system, including the hippocampus, cerebral cortex, cerebellum, amygdala, striatum, retinal bipolar cells, and dorsal root ganglia (DRG) [74,75,80]. KAR subunits have also been found in non-neuronal cells, such as astrocytes and oligodendrocytes, but not in cerebrovascular endothelial cells [81]. GluK2 is the most abundant isoform present in principal cells (e.g., hippocampal and cortical pyramidal cells; hippocampal and cerebellar granule cells; GrCs), whereas GluK1 is more abundantly expressed in hippocampal and cortical interneurons. GluK4 has been mainly detected in the dentate gyrus, CA3 pyramidal neurons, neocortex, and Purkinje cells, whereas GluK5 is abundantly expressed throughout the brain [74,77]. KARs can be located either in the postsynaptic neuron to facilitate neuronal excitability or at the presynaptic terminal to fine-tune neurotransmitter release [74,75]. In accord with this, KARs carry part of the EPSC induced by synaptically released glutamate at certain central synapses, including the mossy fiber (MF)-CA3 synapse and the SchC-CA1 synapse in the hippocampus, the parallel fiber (PF)-Golgi cell synapse in the cerebellum, and at thalamocortical connections [75,77]. KAR-mediated EPSCs (KAR-EPSCs) differ from AMPAR-EPSCs due to their smaller amplitude and slower activation/deactivation kinetics [82]. The slow kinetics of the postsynaptic KAR-EPSCs are likely dependent on Neto proteins [77] and are critical for synaptic integration and synchronization of network activity in the brain [74,75]. By contrast, KAR-dependent regulation of neurotransmitter release and other neuronal functions requires both ionotropic and flux-independent signaling [74,75,77].

### 4.1. Flux-Independent Signaling by Presynaptic KARs Regulates GABA Release in the Hippocampus

Presynaptic KARs can modulate GABA and glutamate release through flux-independent signaling (Figure 2 and Table 1) [74,75,77]. Pharmacological activation of presynaptic KARs with KA reduces the probability of GABA release by CA1 interneurons in the hippocampus through a metabotropic-like mechanism (Figure 2) [83,84,85]. KARs engage a signaling pathway that comprises pertussis toxin (PTx)-sensitive G_i/o_ proteins, PLC, and conventional protein kinase C (cPKC) (Figure 2). The cPKC could then reduce the Ca^2+^ sensitivity of the Ca^2+^ machinery involved in vesicle exocytosis from presynaptic terminals [83,84,86]. Recruitment of cPKC to the PM requires both diacylglycerol (DAG) and Ca^2+^, which is released from the ER by the intracellular second messenger inositol-1,4,5-trisphosphate (InsP_3_) [87]. Notably, both DAG and InsP_3_ are produced by the PLC-dependent hydrolysis of phosphatidylinositol 4,5-bisphosphate (PIP_2_) [88], and PTx-sensitive G_i/o_ proteins can activate PLCβ2 and PLCβ3 through their βγ dimers [89,90,91]. In agreement with this hypothesis, parallel work has shown that KAR activation is able to induce intracellular Ca^2+^ release, which is a proxy for PLC activation [10,92], in mouse DRG neurons (Figure 2) [93]. Ionotropic KARs activated during synaptic activation could facilitate the release of 2-arachidonoylglycerol, which activates presynaptic cannabinoid type 1 (CB1) receptors and inhibits GABA mobilization [94]. However, CB1 inhibitors do not prevent the inhibitory effect of KARs on GABA release [95]. This observation suggests that pharmacologically activated KARs primarily signal the inhibition of GABA mobilization in a flux-independent manner [94]. On the other hand, the endogenous activation of presynaptic KARs stimulates GABA release from hippocampal CA1 interneurons through an ionotropic mechanism that enhances presynaptic depolarization [96,97]. To reconcile these discrepancies, it has been proposed that low concentrations of glutamate stimulate KARs to trigger rapid ionotropic signals that increase GABA release, whereas an increase in agonist concentration induces KARs to switch into the flux-independent signaling mode to reduce GABAergic inhibition [74,98,99,100]. Therefore, by means of their dual signaling mode, presynaptic KARs can sense the level of neuronal activity at individual synapses and trigger the most appropriate intracellular signal to fine-tune synaptic inhibition and maintain network homeostasis [74].

### 4.2. Flux-Independent Signaling by Presynaptic KARs Regulates Glutamate Release in the Hippocampus

Flux-independent signaling by presynaptic KARs can also modulate glutamate release (Table 1) [75]. In the neonatal rat hippocampus, GluK1-containing KARs that are located on MFs (i.e., the axons projected by GrCs in the dentate gyrus) tonically inhibit or promote glutamate release onto CA3 pyramidal cells and GABAergic interneurons, respectively [101]. KARs are activated by ambient glutamate to inhibit glutamate release in a G_i/o_ protein- and cPKC-dependent manner [101]. The G_i/o_ protein βγ dimers could directly inhibit presynaptic Ca_V_2.1 currents, thereby restraining glutamate release [75]. By contrast, the mechanisms by which KARs facilitate glutamate release onto CA3 interneurons remain unclear [101]. Intriguingly, in both CA3 pyramidal cells and GABAergic interneurons, endogenous KAR stimulation enhances GABAergic transmission in a flux-dependent manner [101]. Therefore, presynaptic KARs signal in both an ionotropic and a flux-dependent manner to regulate the synchronous network activity that contributes to rat hippocampus development [75,101]. In fact, the tonic activation of presynaptic KARs in the hippocampal CA3 region is downregulated at two weeks after birth [101]. This may be due to the downregulation of GluK1 expression in mature pyramidal neurons [74].

In the neonatal rat hippocampus, GluK1-containing KARs are also tonically activated by ambient glutamate to inhibit glutamate release from the SchCs projected by CA3 neurons onto the CA1 region (Table 1) [102]. In agreement with this observation, the exogenous administration of KA reduces voltage-dependent Ca^2+^ transients at the SchC [103], thereby suggesting that presynaptic KARs may directly modulate Ca_V_2.1 channels via the recruitment of G_i/o_ proteins (Figure 2) [102]. As observed at the MF-CA3 synapse, the tonic inhibition of glutamate release onto CA1 pyramidal cells is lost during development [102,104]. Physiological activation of presynaptic KARs therefore fine-tunes synapse formation and maturation in the glutamatergic circuitry of the CA1 region in a G_i/o_- and cPKC-dependent manner [105]. Switching the molecular assortment of GluK subunits could reduce the sensitivity of KARs to ambient glutamate (e.g., due to the downregulation of high-affinity GluK4 or GluK5 subunits) and/or decouple the presynaptic receptor from neighboring G_i/o_ proteins in adult rats [104]. However, the pharmacological stimulation of presynaptic KARs at the SchC-CA1 pyramidal neuron synapse can also inhibit glutamate release via G_i/o_ proteins in the fully developed hippocampus [106]. This observation suggests that the flux-independent mechanism that restrains glutamate release in the CA1 region is still at work in the adult hippocampus, where presynaptic KARs are likely to serve as inhibitory auto-receptors to shape short-term plasticity [106]. Interestingly, the inhibitory effect of KARs on glutamate release is prevented by the prior stimulation of adenosine or GABA_B_ receptors, suggesting that classical metabotropic receptors and metabotropic-like KARs converge on the same signaling pathways [107].

Parallel studies have confirmed that presynaptic KARs also regulate glutamate release in the adult mouse hippocampus (Table 1) [75]. It has been suggested that at the MF-CA3 synapse, low nanomolar concentrations of KA (<50 nM) activate KARs to stimulate glutamate release through an ionotropic mechanism that requires the inclusion of the low-affinity GluK2 and the high-affinity GluK5 in the heteromeric channel protein [75,108,109,110]. Another investigation reported that GluK2-/GluK3-containing ionotropic KARs are activated by high concentrations of glutamate to favor synaptic transmission at this same synaptic site [111]. By contrast, high nanomolar concentrations of KA (>100 nM) inhibit glutamate release by stimulating KARs to signal in a flux-independent-mode [75,112,113]. Genetic deletion of GluK5 does not impair this effect, whereas GluK2 is likely involved in both signaling modes [109,114,115]. A recent investigation showed that the GluK2 Q/R editing inhibits the metabotropic-like function in mouse CA3 postsynaptic KARs [116]. Therefore, edited and unedited GluK2 subunits could stimulate and inhibit glutamate release, respectively, the former probably in association with GluK5. Presynaptic KARs inhibit glutamate release at the mouse MF-CA3 synapse by recruiting G_i/o_ proteins to inhibit AC activity, thereby reducing cyclic AMP (cAMP) production and protein kinase A (PKA) activation [112]. This results in reduced exocytosis of neurotransmitter vesicles from the glutamatergic MFs, which is stimulated by PKA-dependent phosphorylation of the secretory machinery [117]. Consistent with this model, Group 2 mGluRs are also coupled to PKA via G_i/o_ proteins [13,18], and their prior inhibition can prevent KA-induced suppression of glutamate release [113]. The evidence that G_o_ proteins are part of the GluK1 interactome [118] strongly suggests that the low-affinity GluK1 is part of the heteromeric KAR protein that negatively regulates glutamate release. It should, however, be pointed out that its role in KAR-mediated glutamate release is highly controversial [115,119].

### 4.3. Flux-Independent Signaling by Presynaptic KARs Regulates Neurotransmitter Release in the Cerebellum

Presynaptic KARs can also modulate neurotransmitter release in a flux-independent manner in the cerebellum (Table 1) [75,80]. 

**Table 1 biomedicines-12-00880-t001:** Modulation of neurotransmitter release by presynaptic KARs signaling in the flux-independent mode.

Brain Region	Effect	Signaling Pathway(s)	Function	Ref.
**Rat hippocampus** CA1 interneurons	GABA ↓	G_i/o_ proteins, PLC, cPKC	Regulation of PN excitability	[83,84,85]
**Rat hippocampus** SChC-CA1 Neonate	Glu ↓	G_i/o_ proteins, cPKC	Synaptic maturation and plasticity	[102,104,105]
**Rat hippocampus** SChC-CA1 Adult	Glu ↓	G_i/o_ proteins, cPKC	Unknown	[106,107]
**Rat hippocampus** MF-CA3 Neonate	Glu ↓ onto PNs	G_i/o_ proteins, cPKC	Hippocampus development	[101]
**Rat hippocampus** MF-CA3 Neonate	Glu ↑ onto GIs	Unknown	Hippocampus development	[101]
**Rat hippocampus** CA3 (A/C)	Glu ↓	G_i/o_ proteins	Vesicle release	[120]
**Mouse hippocampus** MF-CA3 Adult	Glu ↓	G_i/o_ proteins, AC inhibition	Unknown	[112,113]
**Mouse cerebellum** PF-PuC Adult	Glu ↓	G_i/o_ proteins, AC inhibition	Synaptic maturation	[75,80,121]
**Mouse amygdala** MGN-LA Adult	Glu ↓	PKA	Plasticity and oscillations at the theta and gamma bands?	[122]
**Rat globus pallidus**	Glu ↓	G_i/o_ proteins, cPKC	Unknown	[123]

Abbreviations: GIs: A/C: association/commissural; AC: adenylate cyclase; cPKC: conventional protein kinase C; Glu: glutamate; LA: lateral nucleus of the amygdala; MF: mossy fiber; MGN: thalamic medial geniculate nucleus; PF: parallel fiber; PKA: protein kinase A; PNs: pyramidal neurons; PuCs: Purkinje cells; SChC: Schaffer collateral; ↓: reduction; ↑: increase.

PFs are the specialized axons that project from GrCs into the molecular layer. Herein, PFs bifurcate and give raise to T-shaped branches that convey information via excitatory glutamatergic terminals to the dendritic spines of Purkinje cells (PuCs) [38]. Single-channel studies suggested that GrCs express homomeric unedited GluK1 homotetramers, which carry a significant Ca^2+^ permeability [78]. Presynaptic KARs can either facilitate or inhibit synaptic transmission depending on the concentration of the synthetic agonist domoate [75,124]. Furthermore, they can be gated by endogenous glutamate to differentially modulate neurotransmitter release during both low- and high-frequency PF stimulation [124]. Falcón-Moya et al. found that presynaptic KARs facilitate synaptic transmission at low agonist concentrations by mediating Ca^2+^ influx, which in turn triggers Ca^2+^-induced Ca^2+^ release (CICR) from the ER, thereby resulting in a large presynaptic Ca^2+^ transient [80,124,125]. The Ca^2+^ signal then engages calmodulin to stimulate neurotransmitter release via the AC/cAMP/PKA signaling pathway [80,125]. By contrast, at high agonist concentrations, inhibition of glutamate release requires flux-independent signaling: presynaptic KARs inhibit AC via G_i/o_ proteins, and their activity is not impaired by the depletion of the ER Ca^2+^ store [75,121]. In the molecular layer of the cerebellum, presynaptic KARs could function in both a flux-dependent and flux-independent manner to sense low or high agonist concentrations, thereby fine-tuning the consolidation and maturation of the PF-PuC synapse during development [75,80]. Interestingly, presynaptic KARs are involved in LTD induction at the PF-PuC synapse, which is modulated by the paired stimulation of climbing fibers (CFs) [126]. However, the latter lack KARs [75]. Future work will have to assess whether presynaptic KARs contribute to LTD induction by signaling in a flux-independent mode.

In addition to the hippocampus and cerebellum, flux-independent signaling by KARs can also modulate glutamate release in the mouse amygdala and rat globus pallidus, as summarized in Table 1.

### 4.4. Flux-Independent Signaling by Postsynaptic KARs Regulates Neuronal Excitability and Synaptic Plasticity in the Hippocampus

Flux-independent signaling by postsynaptic KARs could regulate neuronal excitability in the hippocampus either by modulating the intracellular Cl^−^ concentration ([Cl^−^]_i_) (Figure 2) or by regulating the after-hyperpolarizing K^+^ current (I_AHP_) (Figure 3).

In mouse CA3 pyramidal neurons, GluK2 can physically interact with the K^+^–Cl^−^ cotransporter 2 (KCC2), which extrudes intracellular Cl^−^ across the PM, thereby lowering the [Cl^−^]_i_ and enabling type A GABA receptor (GABA_A_R)-dependent hyperpolarization and reduction in neuronal excitability (Figure 2A) [74,127,128]. The tight association between GluK2 and KCC2 is likely mediated by Neto2, which binds both proteins [74,127,129]. GluK2 is required to recycle KCC2 from the Rab11-dependent pathway to the PM, but for this to occur, GluK2 must be phosphorylated by cPKC at the COOH-terminal residues S846 and S868 (Figure 2A) [130]. Ion flux through KARs is not involved in KCC2 recycling to the PM, which therefore depends only on direct protein–protein interaction [74,130]. In agreement with these findings, the pharmacological activation of GluK1-/GluK2-containing KARs with KA induced a significant hyperpolarization of the reversal potential for GABA (E_GABA_) and increased the driving force that sustains extracellular Cl^−^ entry, which in turn increases the strength of synaptic inhibition (Figure 2A) [128]. Additionally, the pharmacological activation of KARs with KA stimulates extrasynaptic GABA_A_Rs in CA1 pyramidal neurons, thereby enhancing tonic inhibition and protecting the neurons from over-excitation during intense synaptic activity (Figure 2B) [131].

The I_AHP_ can be activated by a submembrane Ca^2+^ pulse upon the activation of voltage-gated L-type Ca^2+^ channels during the neuronal action potential. The I_AHP_ is carried by big-conductance Ca^2+^-dependent K^+^ channels (BK_Ca_, responsible for the fast after-hyperpolarization, or fAHP: I_fAHP_), small-conductance Ca^2+^-dependent K^+^ channels (SK_Ca_, responsible for the medium AHP, or mAHP: I_mAHP_), and intermediate Ca^2+^-dependent K^+^ channels (IK_Ca_, responsible for the slow AHP, or sAHP: I_sAHP_) [132]. The I_AHP_ shapes the neuronal output during synaptic activation by limiting the firing frequency, leading to spike adaptation [71,132,133]. Low concentrations of KA (200 nM) reduce I_sAHP_ in rat CA1 pyramidal neurons by stimulating KARs to signal in a flux-independent manner: KARs inhibit SK_Ca_ channels, thereby increasing neuronal excitability, via G_i/o_ proteins and cPKC (Figure 3) [134]. A similar signaling mechanism has been reported in the rat CA3 region, where KARs inhibit both I_sAHP_ and I_mAHP_ via cPKC activation [135,136]. The flux-independent regulation of I_sAHP_ and I_mAHP_ in the CA3 region is abolished in transgenic mice lacking GluK2 and GluK5, but not GluK1 [135,136,137]. GluK5 cannot be targeted to the PM in the absence of GluK2 [8], which explains why KA fails to inhibit I_sAHP_ and I_mAHP_ in GluK2-deficient neurons. However, the GluK2 Q/R editing prevents the suppression of I_sAHP_ induced by synaptic MF activation in CA3 pyramidal neurons [116]. Flux-independent signaling by KARs also regulates membrane excitability in CA3 interneurons. In juvenile rats, KARs are tonically activated by ambient glutamate to inhibit I_mAHP_ through G_i/o_ proteins, thereby increasing the interneuron firing rate. During development, KARs are uncoupled from SK_Ca_ channels, and this regulation is lost in the adulthood, thereby strongly reducing interneuron firing [138]. Similarly, flux-independent KAR signaling suppresses I_mAHP_ in immature mouse CA3 interneurons, where GluK2 is targeted to the postsynaptic membrane only in the presence of Neto1 [139].

Repetitive synaptic activation of KARs in mouse CA3 pyramidal neurons leads to a rundown of I_sAHP_ that recovers 10 sec after stimulation [137]. Therefore, a burst of synaptic activation could increase pyramidal cell excitability for a significant period of time via the KAR-dependent suppression of I_sAHP_ [93]. KAR-mediated MF-CA3 synaptic transmission can undergo LTD upon high-frequency MF stimulation, resulting in GluK5 phosphorylation by Ca^2+^/Calmodulin-dependent protein kinase II (CaMKII) [140,141]. LTD of KAR-mediated postsynaptic responses relieves I_sAHP_ inhibition, thereby increasing NMDAR-dependent excitability in response to natural stimulus patterns that mimic GrC activity in vivo [140,141]. Metabotropic-like KARs are also engaged by synaptic activation to reduce I_sAHP_ at the SchC-CA1 synapse, although the rundown at this site was irreversible [94]. Synaptic activation of KARs does not induce EPSCs in CA1 pyramidal cells [94], although they are expressed and functional in the dendritic spines [142]. It has therefore been proposed that ionotropic and metabotropic-like KARs are, at least in part, physically segregated in the postsynaptic compartment of the CA1 pyramidal cell [8], as was originally proposed for DRG cells [93].

Postsynaptic KARs in the mouse CA1 region can also utilize flux-independent signaling to induce NMDAR-independent LTP. High-frequency stimulation of the SchC pathway causes an increase in surface expression of functional AMPARs and in spine size that is not mediated by NMDARs [143]. Synaptically released glutamate stimulates GluK2-containing KARs to trigger the signaling cascade leading to LTP, which involves the recruitment of Rab11-containining endosomes to dendritic spines (Figure 3) [143]. Ionotropic signaling does not contribute to KAR-dependent LTP in CA1 pyramidal neurons. By contrast, postsynaptic KARs activate G_i/o_ proteins, PLC, and PKC. InsP_3_-induced ER Ca^2+^ release in dendritic spines is also required to induce LTP (Figure 3) [143]. Notably, the pharmacological blockade of Group I and Group II mGluRs did not prevent KAR-dependent increase in spine size or synaptic AMPARs [143]. KAR-dependent LTP could also be induced by a more physiological pattern of hippocampal activity, i.e., the sharp-wave/ripple-like stimulation pattern [143]. It is still unclear whether KAR-mediated LTP induction in the hippocampus is facilitated by the I_sAHP_ downregulation. Flux-independent signaling can be activated by KARs to increase neuronal excitability and enable the neuronal ensemble to enter a “learning mode” state in the pyriform cortex (PC) [144], which plays a critical role in the olfactory perception and discrimination. Direct administration of KA or a short tetanic stimulation stimulates KARs to increase the excitability of PC pyramidal neurons by inhibiting I_sAHP_. KARs signal the downregulation of I_sAHP_ by activating cPKC and ERK, whereas synaptic activation fails to increase neuronal excitability in GluK2-deficient transgenic mice and in wild-type mice, in which this signaling cascade is inhibited by successful odor discrimination rule learning [144]. In accord with this, complex odor leaning capability is impaired in GluK2-deficient mice and enhanced by viral-mediated GluK2 overexpression in ex vivo slices [144]. LTP is more readily inducible when the I_AHP_ is reduced [133]. The long-term storage of the olfactory information in the PC may, therefore, be favored by the KAR-mediated suppression of I_sAHP_. However, the odor learning-induced suppression of I_sAHP_ is impaired in PC pyramidal neurons from old mice, suggesting that GluK2 may represent a promising target to alleviate cognitive decline during aging [145]. Taken together, these preliminary findings strongly suggest that flux-independent signaling by KARs is more prone to induce LTP rather than LTD.

### 4.5. Flux-Independent Signaling by KARs Regulates Axon Growth and Synaptic Differentiation

Presynaptic KARs can bidirectionally regulate the rapid motility of axonal filipodia in the mouse hippocampus during development [93]. Low concentrations of KA (1 µM) released from the target CA3 pyramidal cells activate KAR-mediated depolarizing inward currents that stimulate motility by activating voltage-gated Ca^2+^ channels [146]. By contrast, an increase in KA concentration (10 µM) stimulates presynaptic KARs to inhibit filopodial motility through G_i/o_ proteins [146]. It has been proposed that in the early stages of development, the larger distance between the source (CA3 pyramidal cell) and the target (axonal filipodia) dilutes glutamate concentration, thereby facilitating ionotropic signaling by KARs. When filopodia contact the CA3 pyramidal cell and stop moving, the reduction in free extracellular space increases glutamate concentration, which stimulates presynaptic KARs to stabilize the synapse in a flux-independent manner [146].

Similarly, metabotropic-like KARs regulate neurite outgrowth [8]. In mouse DRG neurons, low concentrations of KA (300 nM) stimulate KARs to promote neurite extension via flux-dependent signaling, whereas high concentrations of KA (3 and 10 µM) stimulate KARs to inhibit neurite extension by mediating ion flux [147]. Flux-independent signaling by KARs involves the G_i/o_ protein-dependent activation of cPKC [147], which is likely due to PLC activation [93]. The microtubule-associated Collapsin Response Mediator Proteins 2 and 4 (CMRP2 and CMRP4) are part of the GluK5 interactome [147] and control many processes during neuronal development, including neuronal migration, neuronal polarity, and neurite outgrowth [8]. In addition, cPKC is recruited by flux-independent KAR signaling to phosphorylate at S9 and thereby inhibit the glycogen synthase kinase-3β (GSK-3β) [147]. The cPKC-dependent inactivation of GSK-3β reduces CMRP2 phosphorylation at T514, which suppresses CMRP2-mediated neurite extension [147]. Physical coupling to different signaling cascades may explain why flux-independent signaling by KARs inhibits filopodial motility in the hippocampal CA3 region, whereas it promotes axonal outgrowth in mouse DRG neurons [147].

### 4.6. Flux-Independent Signaling by KARs: Future Perspectives

Most of the reports indicate that metabotropic-like KARs transduce synaptic activation into an intracellular signal via G_i/o_ proteins, PLC, and cPKC. It has been reported that KARs can also interact with G_q_ proteins in a heterologous cell system [137], but this mode of signaling has not yet been reported in a more physiological context. An adaptor protein could connect GluK1-containing KARs to G_i/o_ proteins [115], but this hypothesis also requires further support. Presynaptic KARs primarily modulate neurotransmitter release in developing neuronal networks by signaling in a flux-independent manner. The molecular mechanisms that contribute to shutting down metabotropic-like activity at some synapses, e.g., MF-CA3 and SchC-CA1, in adulthood remain an open question. The molecular underpinnings that enable presynaptic KARs to serve as ionotropic or metabotropic-like receptors of low and high glutamate concentrations, respectively, also have yet to be elucidated. Postsynaptic KARs mainly signal in a flux-independent manner to increase neuronal excitability by suppressing the I_AHP_, thereby potentially favoring LTP induction, as shown in [143]. The requirement for cPKC to inhibit I_sAHP_ and I_mAHP_ suggests that PLC must be activated by KARs via G_i/o_ proteins to generate the PKC agonist, DAG. However, PLC activation always results in InsP_3_-induced ER Ca^2+^ release, which can be evoked by KAR activation in DRG neurons [93] and in CA1 pyramidal neurons [143]. If postsynaptic KARs can activate a Ca^2+^-dependent conductance, such as the I_AHP_, we speculate that the Ca^2+^ signal generated by the concomitant recruitment of ER-embedded InsP_3_Rs is not directed towards the PM, where it would activate the I_sAHP_. Additionally, future investigation will have to assess whether ionotropic and metabotropic-like KARs coexist at the same postsynaptic sites or whether they are physically separated and converge on different signaling outputs, i.e., depolarization and G_i/o_ protein activation, respectively. Finally, unraveling how KARs interact with their protein partners will benefit from the elucidation of their full-length structure. To date, cryo-electron microscopy (cryo-EM) has only captured the structure of either homomeric (GluK2 and GluK3) or heterotetrameric (GluK2/GluK5) KARs in their desensitized state [148]. We envision that capturing the cryo-EM structure of KARs signaling in the flux-independent mode, which is certainly a challenging task, will be useful in understanding how KARs use distinct molecular configurations to utilize different signaling modes.

## 5. Flux-Independent Signaling by NMDARs

NMDARs are heterotetrameric channels containing multiple subunits: GluN1, GluN2A, GluN2B, GluN2C, GluN2D, GluN3A, and GluN3B [15,17]. Two obligatory GluN1 subunits can assemble with either two GluN2 subunits or a combination of GluN2 and GluN3 subunits [9,10]. GluN1 has eight splice variants, although their functional differences and roles remain unclear [149]. Canonical NMDARs consist of two GluN1 subunits, which bind to the NMDAR co-agonists, i.e., glycine or D-serine, and two GluN2 subunits, which bind to the physiological agonist, i.e., glutamate. In addition, GluN1 is necessary for proper assembly and surface delivery of the whole NMDAR protein to the PM [15,17]. Canonical NMDARs mediate the influx of both extracellular Na^+^ and Ca^2+^ and undergo a strong Mg^2+^-dependent inhibition at negative resting potentials. Therefore, NMDARs can serve as coincidence detectors, as their activation requires the simultaneous presynaptic release of glutamate and postsynaptic depolarization, which relieves Mg^2+^ inhibition by extruding Mg^2+^ ions from the channel pore. The extracellular concentration of D-serine (or glycine) is usually sufficient to promote their gating [15,17]. Recent evidence suggests that, at least in certain synapses, D-serine and glycine, respectively, gate synaptic and extrasynaptic NMDARs [150]. An alternative but not mutually exclusive hypothesis is that glycine serves as an NMDAR co-agonist in early developmental stages, whereas D-serine fulfils this role in the adult [150,151,152].

NMDARs are barely activated during the baseline (i.e., low frequency) activity of excitatory glutamatergic synapses, mainly because of the Mg^2+^-dependent inhibition. Conversely, as the frequency of synaptic activation increases, the AMPAR-mediated postsynaptic depolarization is sufficient to repel Mg^2+^ ions from the channel pore, thereby activating the slower NMDAR-EPSCs that last up to hundreds of milliseconds and allow the influx of a substantial amount of Ca^2+^ into the postsynaptic spine [15,17]. GluN2A- and GluN2B-containing NMDARs show a relatively high single-channel conductance (50 pS) and bear a robust permeability for Ca^2+^ (P_Ca_/P_Cs_ ≈ 7) [149]. This high Ca^2+^ permeability enables NMDARs to transduce specific patterns of synaptic activation into long-lasting changes in synaptic strength, such as LTP and LTD [9,15,17,149]. However, the inclusion of GluN2 or GluN2D can significantly impact the biophysical properties of NMDARs, which show reduced single-channel conductance (37 pS), Ca^2+^ permeability (P_Ca_/P_Cs_ ≈ 4.5), and Mg^2+^-dependent inhibition (IC_50_ = 80 µM vs. 35 µM) [9,15,17,149]. The role of GluN3, which also avidly binds glycyne, has only recently been recognized [153,154]. Incorporation of the GluN3A subunit into tri-heteromeric GluN1/GluN2/GluN3 channels further reduces the single-channel conductance and Ca^2+^ permeability and abolishes the Mg^2+^ sensitivity at negative membrane potentials [153,154,155]. Recently, di-heteromeric GluN1/GluN3 channels have been discovered that are not activated by glutamate, as they lack the GluN2 subunit, but are gated by glycine [153]. GluN1/GluN3 channels serve as glycine-gated excitatory receptors that lack Ca^2+^ permeability and Mg^2+^-dependent inhibition and can mediate neurotransmission in some brain regions, including the medial habenula and the juvenile hippocampus [153,154]. This new evidence led to the hypothesis that neuronal NMDARs co-exist as glutamate-gated GluN1/GluN2 and GluN1/GluN2/GluN3 receptors and glycine-gated GluN1/GluN3 [153,154].

Extracellular Ca^2+^ entry through conventional NMDARs plays a primary role in the processes of learning and memory formation due to its ability to modulate the strength of synaptic transmission in both the short and long term [17,156]. However, a flurry of studies have convincingly shown that NMDARs can also signal in a flux-independent manner to regulate a variety of neuronal functions, ranging from neurotransmitter release to synaptic plasticity, whereas excessive metabotropic-like NMDAR signaling is associated with excitotoxicity [5,6,8,24]. Furthermore, flux-independent signaling by GluN1/GluN2 and GluN1/GluN2/GluN3 receptors has also been described in astrocytes [157,158] and cerebrovascular endothelial cells [9,159,160]. This growing body of evidence further expands the versatility of NMDAR signaling at the neurovascular unit.

### 5.1. Flux-Independent Signaling by NMDARs in Neuronal Physiology

Flux-independent signaling by NMDARs regulates several neuronal functions, including NMDAR internalization and trafficking, spine morphology, LTD, and glutamate release (Figure 4).

#### 5.1.1. Flux-Independent Signaling by NMDARs Controls NMDAR Internalization and Trafficking

An early report showed that glutamate binding to NMDARs induces NMDAR internalization in a flux-independent manner with the aid of the clathrin-adaptor protein AP-2 (Figure 4A) [161]. This effect is induced by repeated applications of glutamate (1 mM), requires the presence of glycine, and is dependent on GluN2A dephosphorylation [161]. This mechanism may explain the downregulation of NMDAR currents caused by tyrosine phosphatase activity during whole-cell patch-clamp recordings [162]. A reduction in the surface expression of NMDARs can also be induced by co-agonist binding alone (Figure 4A). High concentrations (10 µM) of glycine prime NMDARs for clathrin-mediated, dynamin-dependent endocytosis, which is induced by subsequent binding to glutamate in isolated rat hippocampal CA1 neurons [163]. The priming effect of glycine does not require ion flux, is mimicked by D-serine, and requires AP-2 (Figure 4A) [163]. Furthermore, glycine-primed internalization of NMDARs does not depend on the GluN2 subunit that interacts with GluN1 but is tightly regulated by a single aminoacidic residue, i.e., A714, in the glycine-binding site [164]. Alternative splicing that removes the N1 polypeptide cassette from the GluN1 subunit is required to gate glycine-induced NMDAR internalization [165]. Thus, glycine priming of NMDARs for endocytosis can only occur in hippocampal neurons, but not in interneurons, which express N1-containing GluN1 subunits [6].

By contrast, spontaneous synaptic activity in the hippocampal CA1 region leads to the incorporation of GluN2A but not GluN2B subunits into postsynaptic NMDARs [166]. Synaptic incorporation of GluN2A requires glutamate and co-agonist binding but is independent of the ion flux through the NMDAR channel pore [166]. This signaling pathway may promote the progressive enrichment of GluN2A-containing NMDARs by adulthood, thereby fine-tuning synaptic maturation, neuronal network formation, and cortical development [167]. The synaptic content of GluN2B-containing NMDARs could be further reduced by D-serine, which decreases the basal surface trafficking of GluN2B, while glycine is ineffective [152]. The D-serine induced reduction in GluN2B trafficking requires a conformational change in the long CTD of GluN2B, which contains several domains involved in the interaction with several postsynaptic density proteins [152], including postsynaptic density-95 (PSD-95) [24,152]. Therefore, fluctuations in the extracellular availability of D-serine (decreases) or glycine (increases) could control the recruitment of GluN2B from the extrasynaptic pool [152], thereby regulating the molecular arrangement of synaptic NMDARs through flux-independent signaling.

#### 5.1.2. Flux-Independent Signaling by NMDARs Controls Synaptic Plasticity

Earlier evidence suggested that hippocampal CA1 neurons may undergo LTD upon low-frequency stimulation of the SchC in the absence of ion flux through NMDARs (Figure 4B) [6,168]. Subsequent investigations confirmed that when NMDAR-mediated Ca^2+^ entry is inhibited either with the selective NMDAR channel pore inhibitor MK-801 or with the co-agonist blocker 7-chlorokynurenic acid (7-CK), NMDAR-mediated LTD can still be induced [169] and lead to spine shrinkage in a Ca^2+^-independent manner [170]. Conversely, LTD is not manifest upon inhibition of glutamate binding with D-2-amino-5-phosphonopentanoate or 3-((RS)-2-carboxypiperazine-4-yl)-propyl-1-phosphonic acid. These findings suggest that NMDARs recruit the signaling pathways that weaken synaptic transmission via a conformational rearrangement that is not associated with Ca^2+^ entry [6,169,170,171] and does not depend on the GluN2 subunit assortment [172]. Intriguingly, blocking flux-dependent signaling with either MK-801 or 7-CK converted high-frequency stimulation-induced LTP into LTD, thereby confirming that NMDARs can signal in a flux-independent manner to reduce synaptic strength [169,170]. Several signaling pathways, including p38 mitogen-activated protein kinase (MAPK) and neuronal NO synthase (nNOS), can sustain the ion flux-independent LTD (Figure 4B) [169,170,173]. The NO synthase 1 adaptor protein (NOS1AP) mediates the interaction between NMDARs and nNOS [173], which is necessary to recruit p38 MAPK to the signaling complex (Figure 4C) [174]. Furthermore, flux-independent NMDAR signaling involves MK2, which is targeted by p38 MAPK to recruit the actin-binding protein cofilin, thereby promoting dendritic spine shrinkage (Figure 4C) [173,175]. CaMKII has long been known to induce LTP but has also been recently implicated in LTD [176]. In accord with this, the pharmacological blockade of CaMKII inhibits the dendritic spine shrinkage induced by flux-independent NMDAR signaling [173]. Förster resonance energy transfer (FRET) imaging revealed that agonist binding to the GluN2 subunit induces a conformational movement of the GluN1 CTDs, which move away from each other and enable protein phosphatase 1 (PP1) to access and dephosphorylate NMDAR-bound CaMKII at T286. PP1-mediated dephosphorylation could reorientate CaMKII within the signaling complex, thereby relocating its catalytic sites in proximity to molecular targets that promote LTD rather than LTP [177]. It has been proposed that dephosphorylated CaMKII, which can be tethered to either GluN1 or GluN2 [177], is responsible for the GluA1 phosphorylation at S567 that occurs during LTD [178,179]. Furthermore, nNOS [180] and CaMKII [176] are Ca^2+^-dependent enzymes, which may explain why buffering basal Ca^2+^ levels prevents NMDAR-mediated LTD [169]. In parallel, basal Ca^2+^ signaling likely stimulates calcineurin to restrain AMPAR expression and reduce AMPAR-EPSCs at the SchC-CA1 synapse [169].

However, other investigations failed to support the emerging evidence of flux-independent NMDAR signaling in the hippocampal CA1 area, as LTD was not induced in the presence of the use-dependent NMDAR channel pore blocker MK-801 [6,178,181]. The possibility of a modest increase in dendritic Ca^2+^ concentration due to interaction between NMDARs and mGluR1/mGluR5, which can occur in cerebrovascular endothelial cells (CECs) [159], was ruled out [170,173]. Flux-independent recruitment of intracellular signaling pathways is mediated by the conformational movement of the GluN1 CTD upon glutamate or NMDA binding to GluN2 [171,177]. It has been shown that an increase in PSD-95 expression reduces NMDA-induced conformational movement in the NMDAR CTD and inhibits flux-independent LTD [182]. Conversely, the overexpression of PSD-95 does not affect NMDAR-mediated, flux-dependent LTD [182,183]. The expression levels of PSD-95 undergo an age-dependent upregulation [184]. Therefore, the PSD-95-dependent obstruction of GluN1 CTD could increase during the first postnatal weeks, thereby promoting flux-independent LTD at more immature synapses [6,182].

Recently, an in vitro investigation showed that the co-agonist glycine can potentiate the AMPAR function by binding to NMDARs containing GluN2A but not GluN2B. This effect occurs in the absence of glutamate and is sensitive to the inhibition of ERK 1/2 while being independent of the ion flux [185]. Interestingly, earlier work has shown that NMDARs stimulate synapse-to-nucleus communication through the Ca^2+^-independent recruitment of the ERK pathway. NMDARs have been shown to synergize with mGluR5 in a PSD-95-dependent manner to induce ERK phosphorylation and promote the expression of the immediate early gene c-fos [186]. Future studies will have to assess whether this mode of metabotropic-like NMDA signaling also occurs in intact hippocampal circuits, thereby replacing the LTD-inducing flux-independent signaling pathways that are inhibited by an increase in PSD-95 expression.

#### 5.1.3. Flux-Independent Signaling by NMDARs Bidirectionally Regulates Spine Morphology

As outlined in Section 5.1.2, flux-independent signaling by NMDARs drives dendritic spine shrinkage following LTD induction at the SchC-CA1 synapse [173,187]. The signaling pathways by which NMDARs drive spine shrinkage are illustrated in Figure 4C and include the mammalian target of rapamycin complex 1 (mTORC1) [188]. The mTORC1 pathway could promote the constitutive protein synthesis that is necessary to induce rapid changes in spine morphology [188]. Alternately, mTORC1 may regulate spine morphology by regulating lysosomal Ca^2+^ release. Synaptic activity redirects lysosomal vesicles into dendritic spines [189], and emerging evidence suggests that lysosomal signaling can exert a bidirectional regulation of synaptic strength through two-pore channel (TPC)-mediated Ca^2+^ release [190]. The mTORC1 phosphorylates TPCs to inhibit local perilysosomal Ca^2+^ signals [191,192]. Flux-independent NMDAR signaling could therefore stimulate mTORC1 to inhibit TPC-mediated Ca^2+^ release, thereby promoting dendritic spine shrinkage and LTD induction.

Surprisingly, a recent investigation showed that flux-independent signaling by NMDARs can also be engaged during LTP induction to promote spine growth (Figure 4C) [193]. The same signaling pathways by which the metabotropic-like activity of NMDARs induces spine shrinkage also support spine growth upon a high-frequency pattern of glutamatergic stimulation: the interaction between NOSP1 and nNOS recruits p38 MAPK into the NMDA signaling complex, followed by MK2-dependent cofilin activation and actin cytoskeleton disassembly (Figure 4C) [193]. However, LTP induction is not affected by the pharmacological blockade of p38 MAPK, whereas Ca^2+^ influx must be associated with the metabotropic-like activity of NMDARs to promote spine growth [193]. It has, therefore, been proposed that flux-independent NMDAR signaling plays a critical role in bidirectional spine structural plasticity: glutamate binding during synaptic activity stimulates NMDARs to engage the p38 MAPK signaling pathway through a conformational modification of their CTD. In the presence of low Ca^2+^ entry, such as during LTD induction, cofilin-induced severing of F-actin leads to dendritic spine shrinkage (Figure 4C). In the presence of strong Ca^2+^ entry, such as during LTP induction, severed G-actin monomers provide the building blocks for CaMKII-dependent F-actin polymerization, extension, and spine growth (Figure 4C) [6,193].

#### 5.1.4. Flux-Independent Signaling by NMDARs Regulates Spontaneous Glutamate Release

Flux-independent NMDAR signaling regulates spontaneous glutamate release both in hippocampal CA1 pyramidal neurons [194] and at the excitatory connections onto layer 5 pyramidal neurons in the visual cortex [195]. In the hippocampus, postsynaptic NMDARs inhibit spontaneous glutamate release via flux-independent trans-synaptic signaling. NMDARs are gated via the simultaneous binding of glutamate and the co-agonist to initiate Src-mediated phosphorylation of pannexin-1 (PANX-1), which mediates anandamide (AEA) entry into the postsynaptic neuron, thereby promoting AEA degradation via the fatty acid amid hydrolase (FAAH) (Figure 4D) [194]. AEA can serve as an endogenous ligand for the non-selective cation channel Transient Receptor Potential Vanilloid 1 (TRPV1), which mediates Ca^2+^ entry [196,197] and regulates the spontaneous exocytosis of glutamate vesicles [194]. The clearance of synaptic AEA prevents TRPV1-mediated Ca^2+^ entry into the presynaptic terminal and reduces spontaneous glutamate release (Figure 4D) [194]. By contrast, presynaptic NMDARs stimulate glutamate release during evoked activity through Ca^2+^ influx, whereas they support spontaneous release via a C-Jun N-terminal kinase-2 (JNK2)-dependent mechanism that is insensitive to Mg^2+^-dependent inhibition (Figure 4D) [195]. It remains to be elucidated whether ionotropic and flux-independent NMDARs are located at different presynaptic sites to control the release of different synaptic pools. This hypothesis is supported by the evidence that the Rab3-interacting molecules (RIM) RIMα and RIMβ, which function as both scaffolding and signaling proteins at the release sites [198], are required by ionotropic but not flux-independent NMDARs [195].

#### 5.1.5. Flux-Independent Signaling by NMDARs in Neuronal Physiology: Future Perspectives

The discovery that the low-frequency glutamatergic stimulation of NMDARs can induce LTD even when the ion flux through the channel pore is inhibited argued against the role of a weak increase in dendritic Ca^2+^ concentration [6,169,177,179]. However, other investigations showed that the use-dependent NMDAR open channel pore blocker MK-801 prevented LTD in the hippocampal CA1 area [178,181,199]. As outlined in Section 5.1.2, an increase in PSD-95 expression during the first postnatal weeks prevents the conformational modifications that occur in the GluN1 CTD1 upon glutamate binding to GluN2. Therefore, LTD induction may require flux-independent NMDAR signaling in younger animals, whereas it relies on NMDAR-mediated extracellular Ca^2+^ entry in older mice [6]. Future studies are mandatory to resolve this discrepancy and should therefore be carried out under more homogenous recording conditions, taking into account the multiple variables that could have led to these contrasting findings, including methods of hippocampal slice preparation, LTD induction protocols, MK-801 treatment, intracellular and extracellular solution, etc. [6]. It would also be worthwhile to investigate whether flux-independent NMDAR-mediated LTD can occur in other brain regions, such as the cortex, amygdala, and cerebellum [72,200,201].

The conformational movements of the GluN1 CTD also deserve careful investigation, as this long cytosolic domain is critical for the interaction with the signaling proteins recruited upon glutamate binding. Interestingly, while agonist binding causes the GluN1 CTDs to move away from each other independently of the ion flux [171,177], FRET fluorescence lifetime imaging showed that the co-agonist D-serine causes the GluN1 CTDs to move closer to each other [152]. By contrast, the other co-agonist, i.e., glycine, does not induce any FRET-detectable movement in the GluN1 CTDs [152], although it could cause movements of the GluN2 CTD that have not yet been captured [6]. D-serine-induced conformational rearrangement of the GluN1 CTD is required to fine-tune NMDAR trafficking and localization to the postsynaptic density. Notably, glutamate causes the GluN1 CTDs to move away from each other both in the presence and in the absence of the co-agonist [152,171]. Therefore, future work could attempt to shed light on the reason why D-serine does not counterbalance GluN1 CTD movements during agonist-induced flux-independent NMDAR signaling.

### 5.2. Flux-Independent Signaling by NMDARs in Brain Astrocytes and Microvascular Endothelial Cells

The neurovascular unit (NVU) represents the smallest functional unit of the brain and fine-tunes synaptic activity and plasticity, neuronal metabolism, local cerebral blood flow (CBF), and the integrity of the blood–brain barrier (BBB) [51,202,203]. At the NVU, NMDARs have also been found in astrocytes [204], vascular smooth muscle cells (VSMCs) [205], and microvascular endothelial cells [9]. Ionotropic NMDARs sense synaptically released glutamate to regulate heterosynaptic plasticity in astrocytes [206] and local changes in CBF in both arteriolar VSMCs [205] and CECs [207]. However, recent findings have revealed that NMDARs can also signal in a flux-independent manner in both rat cortical astrocytes [157,158] and human brain microvascular endothelial cells [159,160].

#### 5.2.1. Flux-Independent Signaling by NMDARs in Brain Astrocytes

Astrocytic NMDARs consist of heteromeric trimers of two GluN1 subunits, one GluN2C or GluN2D subunit and one GluN3 subunit [204,208]. The role of ionotropic NMDARs in brain astrocytes is still controversial [209], although preliminary evidence suggests that they can be activated during high-frequency stimulation of the SchC-CA1 synapse to induce LTD at adjacent non-stimulated inputs [206]. Nevertheless, several recent studies have shown that NMDARs signal an increase in [Ca^2+^]_i_ in a flux-independent manner in rat cortical astrocytes. NMDA was first found to induce Ca^2+^ release from the ER, which was inhibited by the selective inhibition of GluN2B with ifenprodil but still occurred in the absence of extracellular Ca^2+^ (0Ca^2+^) [210]. NMDA-evoked Ca^2+^ signals are dose-dependent: they arise at 0.1 nM and reach a maximum amplitude at 100 nM [210]. The intracellular Ca^2+^ response to NMDA is shaped by InsP_3_-induced ER Ca^2+^ release through InsP_3_Rs) [210]. Type 2 InsP_3_Rs (InsP_3_R2) are the major ER Ca^2+^-releasing channels in astrocytes [211], which do not express ryanodine receptors that can amplify InsP_3_-evoked Ca^2+^ release via CICR in neurons [212]. The amplitude of the Ca^2+^ response is reduced under 0Ca^2+^ conditions, whereas its duration is curtailed [210]. The Ca^2+^ entry pathway responsible for NMDA-induced Ca^2+^ entry has not been investigated, but it could be provided by SOCE [213]. SOCE is the main Ca^2+^ source for ER Ca^2+^ refilling in astrocytes upon InsP_3_-induced depletion of the ER Ca^2+^ store [49,214]. In addition, the interplay between STIM proteins and GluA1/GluA2 subunits may also contribute to astrocytic SOCE [54]. Future work could investigate whether InsP_3_-induced ER Ca^2+^ release sets in motion an unexpected communication between NMDARs and AMPARs in astrocytes. A subsequent report confirmed that NMDA induces a dose-dependent (1–100 µM) delayed and sustained increase in [Ca^2+^]_i_ in rat cortical neurons that is initiated by InsP_3_-induced ER Ca^2+^ release and maintained via Ca^2+^ entry [215]. The kinetics of the long-lasting influx of Ca^2+^ induced by the prolonged application of NMDA are again consistent with SOCE activation [50], but this hypothesis remains to be supported by experimental data. NMDA-evoked Ca^2+^ signaling recruits an antioxidant defense program in rat cortical astrocytes: the increase in [Ca^2+^]_i_ recruits protein kinase Cδ to phosphorylate and stabilize p35, which is a cyclin-dependent kinase-5 (Cdk5) cofactor [215]. The active p35/Cdk5 complex promotes the nuclear translocation of Nrf2 through phosphorylation of T395, S433, and T439, thereby inducing the expression of several antioxidant genes [215]. Furthermore, the NMDARs–Cdk5–Nrf2 signaling pathway promotes the release of glutathione (GSH) precursors, leading to an increase in neuronal GSH in co-culture. Astrocytes-derived GSH, in turn, affords protection against oxidative stress to neighboring neurons [215]. The InsP_3_-dependent astrocytic Ca^2+^ response to NMDA was confirmed by a third independent investigation showing that the Ca^2+^ signal was unaffected by MK-801, whereas it was suppressed by the genetic deletion of GluN1 [157]. This response is elicited by a rather high NMDA concentration (1 mM) that may be within the range of glutamate levels reported during high neuronal activity at glutamatergic synapses [24,157]. This rather high NMDA concentration results in acidification of the extracellular milieu that reaches pH ≈ 6.0 [157]. A follow-up study revealed that NMDARs can serve as pH sensors in rat cortical astrocytes and that the Ca^2+^ signal does not occur when 1 mM NMDA is administered through an extracellular solution buffered at a pH of 7.0 [158]. Therefore, astrocytic NMDARs may play a critical role during intense synaptic activity and may also be activated by pathological conditions that lead to acidification of the extracellular milieu, such as hypercapnia, inflammation, and stroke [157,158].

Future work is required to assess whether flux-independent NMDAR signaling elicits intracellular Ca^2+^ waves or any other signaling pathways in astrocytes from other brain regions. In addition, the contribution of the lysosomal Ca^2+^ pool, which triggers the InsP_3_-evoked Ca^2+^ pulse in neurons, astrocytes, and CECs, to NMDAR-dependent ER Ca^2+^ release should also be assessed [216]. The interaction between NMDARs and STIM-regulated proteins also requires careful consideration [48,49]. On the one hand, the ER resident STIM proteins could sense the decrease in ER Ca^2+^ concentration induced by NMDA-induced Ca^2+^ release through InsP_3_Rs, as outlined above. On the other hand, STIM activation by flux-independent NMDAR signaling could promote GluN1, GluN2A, and GluN2B endocytosis, as reported in rat cortical neurons [217,218]. Current evidence suggests that NMDAR-mediated currents are only detectable in vivo and not in cultured astrocytes [204]. It has been proposed that a signaling switch occurs in cortical astrocytes maintained in culture [204]. STIM-dependent endocytosis may be one of the mechanisms contributing to suppress NMDA-evoked non-selective cation currents in primary astrocyte cultures. Therefore, assessing whether flux-independent NMDAR signaling occurs only in culture or can also be activated in vivo is essential to understand its potential physio-pathological relevance.

#### 5.2.2. Flux-Independent Signaling by NMDARs in Human CECs

Endothelial cells line the inner lumen of blood vessels and are therefore critical for regulating the exchange of solutes between the streaming blood and parenchymal tissues [35,219]. The primary role of CECs has long been associated with the maintenance of the structural and functional integrity of the BBB [220]. The BBB is composed of capillary endothelial cells, pericytes, and astrocyte end-feet that cooperate to supply neurons with oxygen and nutrients and to remove their catabolic waste [221]. Recent studies have provided further insight into the physiological function of CECs, which have also been shown to trigger neurovascular coupling (NVC) by translating the local increase in cortical activity into an increase in local CBF [88,222]. Several ionic signaling mechanisms enable CECs to sense neuronal activity [88,222], including NMDARs, which are normally located in the basolateral membrane, i.e., in the most suitable position to sense synaptically released glutamate [223]. Studies conducted in mouse brain microvasculature have shown that in pre-capillary arterioles, endothelial NMDARs increase local CBF by stimulating the Ca^2+^-dependent endothelial NO synthase [223,224], whereas they promote an increase in BBB permeability at the capillary level [225,226]. Interestingly, NMDARs have also been shown to signal in a flux-independent manner in human CECs [9].

Primary CECs isolated from human microvessels express GluN1 [226,227], whereas the hCMEC/D3 cell line, which is the most widely employed model of human BBB [228,229], expresses GluN1, GluN2C, and GluN3B subunits [159]. Assembly of GluN1 with GluN2 and GluN3B would reduce the single-channel conductance, Ca^2+^ permeability, and sensitivity to extracellular Mg^2+^ of the resulting NMDAR [9,230]. Electrophysiological recordings showed that NMDA failed to elicit inward currents in hCMEC/D3 cells either in the absence or in the presence of extracellular Mg^2+^, nor could currents be gated in the presence of the co-agonist D-serine [159]. However, NMDA induced a robust increase in NO levels that was abolished by the genetic deletion of GluN1, by the removal of extracellular Ca^2+^, and by the blockade of the NMDAR channel pore with MK-801 [159]. These findings strongly suggest that the human NMDAR is a hetero-trimeric channel consisting of (1) two GluN1, i.e., the obligatory subunit for channel assembly; (2) one GluN2C, which is required for agonist binding; and (3) GluN3C, which is responsible for a low-amplitude current that falls below the resolution limit of whole-cell amplifiers and weak Ca^2+^ permeability that results in a spatially restricted submembrane Ca^2+^ microdomain [9,159]. Consistent with this, single-cell Ca^2+^ imaging revealed that NMDA induces a dose-dependent (3–300 µM) biphasic increase in [Ca^2+^]_i_ that was only curtailed but not abolished in the absence of extracellular Ca^2+^. Moreover, the Ca^2+^ response to NMDA was strongly reduced by the genetic deletion of GluN1 and by the pharmacological blockade of agonist binding with 2-amino-5-phosphonopentanoic acid APV, whereas it was unaffected by MK-801. Therefore, extracellular Ca^2+^ entry through the channel pore is not sufficient to trigger a global Ca^2+^ wave, and NMDARs must signal the increase in [Ca^2+^]_i_ in a flux-independent manner [159]. In agreement with this hypothesis, the intracellular Ca^2+^ response to NMDA was abolished by preventing ER Ca^2+^ release through InsP_3_Rs and lysosomal Ca^2+^ mobilization through TPCs [159]. In the endothelial lineage, nicotinic acid dinucleotide phosphate (NAADP) gates lysosomal TPCs to elicit local Ca^2+^ release that is then amplified by juxtaposed InsP_3_Rs through the CICR process, thereby triggering the Ca^2+^ response to extracellular autacoids [231,232,233]. In contrast to astrocytic NMDARs, strong evidence has been provided for SOCE as the major Ca^2+^ entry pathway that sustains Ca^2+^ entry during flux-independent NMDAR signaling in hCMEC/D3 cells [159]. Furthermore, the metabotropic Ca^2+^ response to NMDA was suppressed by the pharmacological and genetic blockade of mGluR1 and mGluR5 [159]. This finding strongly suggests that NMDARs are either functionally or structurally coupled with Group 1 mGluRs, which could induce the synthesis of InsP_3_ and NAADP by recruiting PLCβ and Dual NADPH oxidase 2 (DUOX2), respectively [10,234]. This hypothesis is supported by the evidence that ERK activation in rat striatal neurons requires flux-independent NMDAR signaling and mGluR5 stimulation [186], as outlined in Section 5.1.2. Furthermore, NMDARs and mGluR5 are closely associated with regards to postsynaptic density [235,236]. The vasorelaxing gasotransmitter NO also plays a crucial role in NVC in human brain microvasculature, although the underlying NOS isoform(s) remain(s) to be identified [237,238]. Ionotropic NMDARs may drive a fast hemodynamic response that follows the onset of neuronal activity, whereas the flux-independent prolonged increase in [Ca^2+^]_i_ signal may maintain NO release during sustained stimulation.

A parallel investigation confirmed that extracellular Ca^2+^ entry through NMDARs in hCMEC/D3 cells is not detectable via Ca^2+^ imaging [160]. However, NMDAR activation via the co-application of NMDA and tissue-type plasminogen activator (tPA) or glycine and tPA recruits the RhoA-/Rho-associated protein kinase (ROCK) signaling pathway; ROCK, in turn, induces the myosin light chain (MLC) kinase (MLCK)-dependent phosphorylation of MLC, thereby increasing hCMEC/D3 cell permeability [239]. The underlying remodeling of the actin cytoskeleton could also be influenced by the NMDAR-mediated increase in [Ca^2+^]_i_ [240], but this hypothesis requires further support.

Endothelial NMDARs have been reported to function only as ionotropic receptors in mouse CECs [9,230]. However, the Anderson group suggested that NMDARs may elicit NO release in response to neural activity through a non-ionotropic mechanism [224]. Mouse brain microcirculation has long been investigated to gain cellular and molecular mechanisms that regulate the functions subserved by the human BBB [51,220,240,241]. However, emerging evidence suggests that the blend of ion channels by which CECs detect neuronal activity can differ between mouse and human microvasculature [242,243,244]. Future work could assess whether NMDARs also signal in a flux-independent manner in mouse brain microvascular endothelial cells.

## 6. Flux-Independent Signaling by NMDARs in Brain Disorders

It has long been known that excessive Ca^2+^ entry through NMDARs can result in Ca^2+^ overload and thereby induce neuronal excitotoxicity, which can manifest as either rapid lytic death (necrosis) or slower programmed cell death (apoptosis) [245,246]. NMDAR hyperactivation can occur upon massive glutamate release, e.g., during epilepsy or acute brain ischemia, or when neurons experience metabolic or oxidative stress, e.g., in neurodegenerative disorders or after traumatic brain injury [245,246]. However, recent studies showed that flux-independent NMDAR signaling can also lead to neuronal dysfunction and excitotoxicity [6].

### 6.1. Neuronal Excitotoxicity

Early work showed that high concentrations of NMDA (200 µM) induce an increase in [Ca^2+^]_i_ in mouse cortical neurons that requires D-serine but is affected by neither the removal of extracellular Ca^2+^ nor by extracellular Mg^2+^ (Figure 5A) [247]. Conversely, the Ca^2+^ response to NMDA is prevented by prior depletion of the ER Ca^2+^ pool [247], suggesting that NMDARs can also recruit ER-embedded InsP_3_Rs in neurons (Figure 5A). ER Ca^2+^ release through InsP_3_Rs promotes NMDAR-mediated phosphorylation of eukaryotic Elongation Factor 2 (eEF-2), which in turn dampens protein synthesis (Figure 5A) [247]. This observation may help us explain how excitotoxic glutamatergic stimulations lead to the inhibition of protein synthesis during brain ischemia. A subsequent investigation confirmed that flux-independent signaling by NMDARs can stimulate pro-death signaling pathways during stroke [248]. Furthermore, NMDAR-dependent, Src-mediated phosphorylation of PANX1 results in robust inward currents and Ca^2+^ entry through PANX1 channels in mouse hippocampal neurons (Figure 5A). This Ca^2+^ influx drives the mitochondrial permeability transition pore (mPTP) formation and neuronal apoptosis during oxygen and glucose deprivation in vitro (Figure 5A). PANX1 is phosphorylated by Src kinase at Y308, which is located on its COOH-terminal tail. Preventing PANX1 phosphorylation with TAT-Panx_308_, an interfering peptide that mimics the COOH-terminal epitope of PANX1, reduces the size of brain lesions and rescues sensorimotor deficits after a stroke in mice [248]. Finally, high concentrations of NMDA (100 µM) can lead to excessive reactive oxygen species (ROS) production by stimulating flux-independent NMDAR signaling in mouse cortical neurons (Figure 5A) [249]. Glutamate binding alone promotes the interaction between the COOH-terminal domain of GluN2B and the p85 regulatory subunit of PI3K (Figure 5A). This enables PI3K to stimulate ROS production by neuronal NADPH oxidase-2 (NOX2) [249], which may occur via the phosphorylation of the NOX2 ^p47^phox subunit at S328 [250]. However, NMDAR-mediated NOX2 activation also requires an increase in [Ca^2+^]_i_, although the source of this Ca^2+^ signal (e.g., ionotropic NMDARs or voltage-gated Ca^2+^ channels) does not seem to be relevant [249]. The excessive increase in intracellular ROS levels induces NMDAR-dependent excitotoxic DNA damage, lipid peroxidation, and neuronal cell death [249].

These findings strongly support the notion that flux-independent NMDAR signaling may exacerbate or even trigger brain damage and loss of cognitive functions after stroke. In agreement with this hypothesis, clinical trials showed that blocking the NMDAR channel pore with either aptiganel (Cerestat) [251] or Mg^2+^ (the FAST-Mag trial) [252] does not afford significant neuroprotection to stroke patients. Therefore, the flux-independent signaling pathway recruited via agonist and co-agonist binding to NMDAR may represent an alternative molecular target for neuroprotective strategies.

### 6.2. Alzheimer’s Disease

One of the major hallmarks of Alzheimer’s disease is the excessive production of the amyloid β (Aβ) protein, which can accumulate to form extracellular amyloid deposits and thereby impair neuronal activity and synaptic transmission in the brain [253]. Two independent studies showed that the Aβ protein-induced depression of synaptic activity at the hippocampal SchC-CA1 synapse depends on flux-independent signaling by GluN2B-containing NMDARs (Figure 5B) [254,255]. The Aβ protein-induced synaptic deficit requires agonist binding to GluN2B and is driven by AMPAR endocytosis and removal from the PM (Figure 5B) [254,255]. A subsequent investigation demonstrated that Aβ oligomers stimulate flux-independent NMDAR signaling to increase p38 MAPK activity and promote dendrite spine shrinkage (Figure 5B), which is a hallmark of synaptic loss and cognitive decline [256]. The evidence that flux-independent NMDAR signaling can lead to LTD (see Section 5.1.2) and to Aβ protein-induced synaptic loss via p38 MAPK recruitment suggests that the synaptic dysfunction associated with AD may involve non-canonical NMDAR activation [6]. This hypothesis was supported by the finding that Aβ protein induces a conformational change in the GluN1 CTD that mimics that induced by agonist binding to GluN2B in the absence of the co-agonist [257]. In addition, as reported for LTD induction in Section 5.1.2, PSD-95 overexpression prevents Aβ-induced conformational change in the CTD and synapse weakening at the SChC-CA1 synapse [257]. These findings suggest that the pharmacological blockade of PSD-95 depalmitoylation, which maintains PSD-95 at the PSD, is a promising strategy to prevent cognitive decline in AD patients [6].

### 6.3. Schizophrenia

Schizophrenia has been associated with a decrease in cerebrospinal fluid D-serine levels and an increase in kynurenic acid, which is an endogenous blocker of the co-agonist binding site of NMDARs [6]. These conditions may favor agonist binding in the absence of the co-agonist [6], which has been shown to induce flux-independent elimination of dendritic spines [170] and synaptic weakening [169]. A recent investigation found that in a mouse model of schizophrenia lacking serine racemase (i.e., the enzyme responsible for D-serine production), high-frequency stimulation of CA1 pyramidal neurons results in spine shrinkage due to flux-independent NMDAR signaling [258]. It has been proposed that in the absence of the co-agonist, a larger fraction of postsynaptic NMDARs signal in a flux-independent manner, whereas ionotropic Ca^2+^ entry is reduced and is not sufficient to drive CaMKII-dependent spine growth [258]. This preliminary finding suggests that NMDAR hypofunction caused by lower D-serine levels may exacerbate flux-independent NMDAR signaling, thereby driving a bias towards spine shrinkage and contributing to the reported reduction in spine density associated with schizophrenia [6,258].

## 7. Conclusions

Evidence from the last three decades supports the notion that the dogmatic classification of postsynaptic glutamate receptors into iGluRs and mGluRs should be replaced by a more flexible interpretation of their signaling activities. A plethora of in vitro and ex vivo studies revealed that AMPARs, KARs, and NMDARs can signal in both a flux-dependent and flux-independent manner. This versatility expands the repertoire of functions that are regulated by iGluRs, which can exert opposing effects on the same neuronal processes, depending on whether they carry an inward ionic current or activate an ion-independent signaling cascade. This dual role of iGluRs is made clear by the requirement for ionotropic or non-ionotropic NMDARs to induce LTD or LTP, respectively. Nevertheless, this novel facet of iGluR signaling is fully consistent with the emerging concept of multifunctional or pleiotropic proteins, which regulate multiple functions through the distinct signaling modules of their quaternary structure. Understanding the molecular determinants that enable iGluR to switch from one signaling mode to the other, which may also depend on the local microenvironment (e.g., D-serine or glycine) or the concomitant activation of other signaling pathways (e.g., Ca^2+^ entry/release channels), is paramount to understanding how they regulate brain functions. Furthermore, this novel signaling mode may provide an unexpected target to design more effective neuroprotective strategies aimed not only at blocking ion flux but also at preventing the agonist-induced conformational rearrangement of cytosolic domains.

## Figures and Tables

**Figure 1 biomedicines-12-00880-f001:**
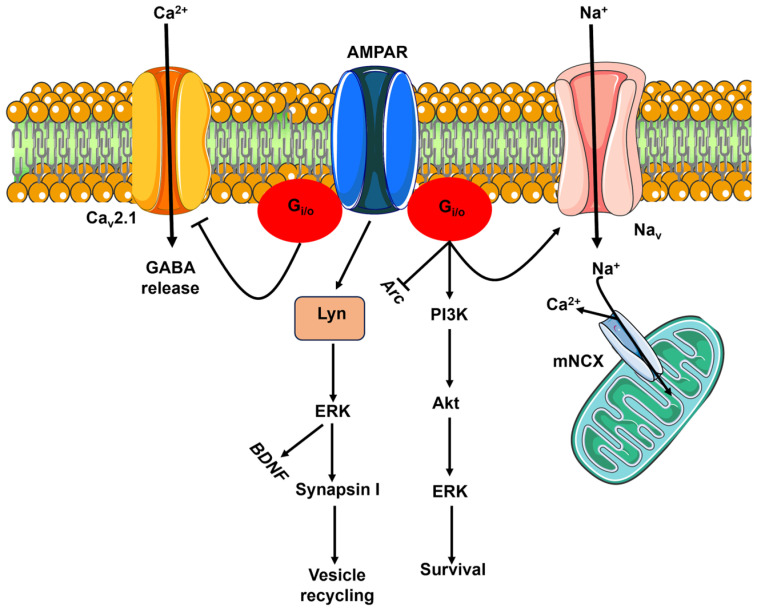
Flux-independent signaling pathways activated by AMPARs in neurons. AMPARs can signal in a flux-independent manner to recruit several signaling pathways. AMPARs can interact with G_i/o_ proteins to either inhibit voltage-gated Ca_V_2.1 channels or activate voltage-gated Na^+^ channels (Na_V_). Extracellular Na^+^ entry through Na_V_ channels can promote mitochondrial Ca^2+^ release through the mitochondrial Na^+^/Ca^2+^ exchanger (mNCX). In addition, AMPARs can signal through G_i/o_ proteins to accelerate vesicle recycling or induce *BDNF* gene expression through ERK activation, which is mediated by the interaction between AMPARs and the tyrosine kinase, Lyn. Alternately, AMPARs can activate the ERK phosphorylation cascade to promote cell survival via the G_i/o_ protein-dependent recruitment of the PI3K/Akt pathway. Finally, AMPARs can also signal through G_i/o_ proteins to inhibit the expression of the *Arc* gene.

**Figure 2 biomedicines-12-00880-f002:**
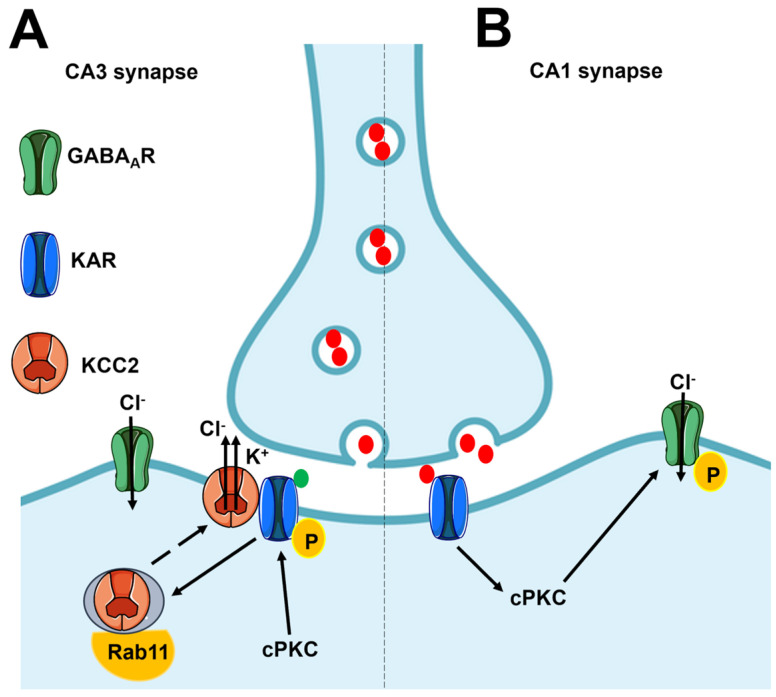
Flux-independent signaling by KARs modulates the intracellular Cl^−^ concentration ([Cl^−^]_i_) and the reversal potential for GABA (E_GABA_). (**A**) Pharmacological stimulation of postsynaptic KARs with KA (green circle) reduces the [Cl^−^]_i_ in hippocampal CA3 pyramidal neurons. Upon permissive cPKC-dependent phosphorylation, KARs promote the recycling of the K^+^–Cl^−^ cotransporter 2 (KCC2) from Rab11-containing vesicles to the PM; the tight interaction between KAR and KCC2 stimulates K^+^ and Cl^−^ efflux into the extracellular milieu, thereby reducing the [Cl^−^]_i_ and increasing extracellular Cl^−^ influx through GABA_A_Rs. (**B**) Synaptically released glutamate (red circle) stimulates flux-independent signaling by KARs, which engage cPKC to phosphorylate extrasynaptic GABA_A_Rs and thereby increase extracellular Cl^−^ influx in hippocampal CA1 pyramidal neurons.

**Figure 3 biomedicines-12-00880-f003:**
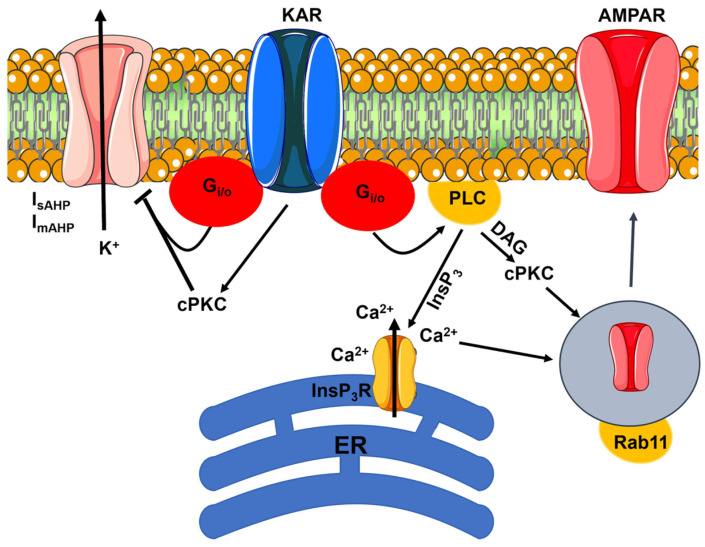
Flux-independent signaling by KARs modulates neuronal excitability and synaptic plasticity. Flux-independent signaling by KAR can increase neuronal excitability by inhibiting I_sAHP_ and I_mAHP_ via G_i/o_ proteins and cPKC. Additionally, KARs can stimulate the exocytosis of AMPARs from Rab1-containing vesicles on the PM of dendritic spines. KARs trigger a signaling cascade involving G_i/o_ proteins and phospholipase C (PLC). PLC, in turn, synthesizes the two intracellular second messengers: DAG, which activates cPKC, and InsP_3_, which induces ER Ca^2+^ release through InsP_3_ receptors (InsP_3_Rs). The combined effect of cPKC and Ca^2+^ release results in the recruitment of Rab11 endosomes to the PM, thereby increasing the surface expression of AMPARs.

**Figure 4 biomedicines-12-00880-f004:**
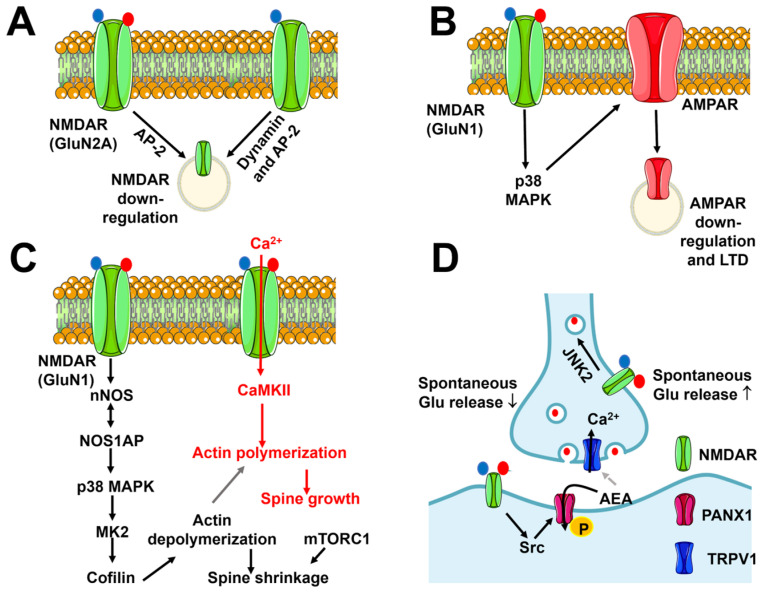
Flux-independent signaling by NMDARs in neurons. (**A**) Agonist and co-agonist binding leads to the dephosphorylation of GluN1 Y837 and GluN2A Y842, resulting in clathrin-dependent internalization via the adaptor protein AP-2. High concentrations (10 µM) of glycine alone increase the interaction with AP-2, thereby priming NMDARs for dynamin-dependent endocytosis upon agonist binding. (**B**) Agonist and co-agonist binding to GluN1-containing NMDARs can stimulate the p38 MAPK in a flux-independent manner to promote AMPAR endocytosis, resulting in Ca^2+^-independent LTD induction. (**C**) Flux-independent signaling by NMDARs regulates spine morphology. In the presence of weak Ca^2+^ entry, NMDARs promote dendritic spine shrinkage by triggering a signaling pathway that requires interaction between nNOS and NOS1P, involving p38 MAPK, MK2, and cofilin, which promotes actin depolymerization. This signaling pathway is supported by mTORC1, which is likely to drive new protein synthesis. In the presence of strong Ca^2+^ influx (highlighted in red), the Ca^2+^-dependent recruitment of CaMKII leads to dendritic spine growth via inducing actin polymerization. (**D**) Flux-independent signaling by postsynaptic NMDARs regulates glutamate release by stimulating Src kinase to activate pannexin 1 (PANX1) channels, which clear synaptic anandamide (AEA) and prevent AEA-induced activation of presynaptic TRPV1 channels. This causes a reduction in [Ca^2+^]_i_ at the presynaptic terminal and therefore decreases Ca^2+^-dependent glutamate release. By contrast, upon agonist and co-agonist binding, presynaptic NMDARs promote glutamate release via the JNK2-dependent signaling pathway. In all the panels, the red circle indicates the agonist, whereas the blue circle indicates the co-agonist. ↑: increase.

**Figure 5 biomedicines-12-00880-f005:**
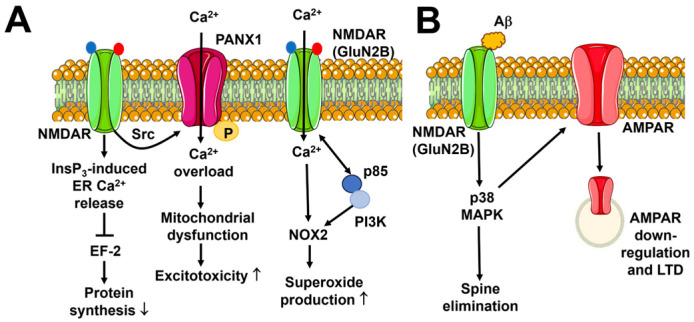
Flux-independent signaling by NMDARs in brain disorders. (**A**) Flux-independent signaling by NMDARs leads to neuronal excitotoxicity upon massive glutamatergic stimulation. NMDARs trigger InsP_3_-induced ER Ca^2+^ release, which inhibits the EF-2 protein and interferes with protein synthesis. Furthermore, NMDARs can stimulate Ca^2+^ entry through PANX1 channels via Src-dependent phosphorylation of PANX1. Excessive Ca^2+^ entry leads to mitochondrial Ca^2+^ overload, mPTP opening, and apoptosis. Finally, agonist binding to GluN2B-containing NMDARs can remove the p85 regulatory subunit from the catalytic domain of PI3K, thereby inducing PI3K-dependent NADPH oxidase-2 (NOX2) activation. NOX2 can also be activated via Ca^2+^ entry through ionotropic NMDARs and lead to cytotoxic superoxide production. The red circle indicates the agonist, whereas the blue circle indicates the co-agonist. (**B**) The Aβ protein binds to GluN2B-containing NMDARs to induce a flux-independent signaling pathway that leads to AMPAR removal from the PM and dendritic spine shrinkage via the p38 MAPK signaling pathway. ↓: decrease; ↑: increase.

## Data Availability

Not applicable.

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
