# Peer review of "Two Signaling Modes Are Better than One: Flux-Independent Signaling by Ionotropic Glutamate Receptors Is Coming of Age"

_biomedicines, 2024, doi:10.3390/biomedicines12040880_

Round 1

Reviewer 1 Report

Comments and Suggestions for Authors

Submitted manuscript is a very interesting and comprehensive review on an interesting aspects of ionotropic glutamate receptors and, in my opinion, it will be of a great interest to readers.

Author Response

We do thank the Reviewer for her/his nice comments to our work. Thank you!!!

Reviewer 2 Report

Comments and Suggestions for Authors

The manuscript written by authors is well explained. I have only minor suggestions.

At few places there are typographic and syntactic mistakes that should address the authors to a careful and deep revision of the manuscript.

The figure legends and tables footnotes need improvement. All legends/table footnotes should have enough description for a reader to understand the figure/tables without having to refer back to the main text of the manuscript.

Many statements in the text are not duly referenced. Latest references should have been consulted and referred. Please Note.

Several unnecessary abbreviations have been used; Author could use the abbreviated forms of words which are repeatedly used in the manuscript.

All acronyms for national agencies, examinations, etc., should be spelled out the first time they are introduced in text or references. Thereafter the acronym can be used if appropriate, e.g. “The work of the Assessment of Performance Unit (APU) in the early 1980s …” and subsequently, “The APU studies of achievement …”, in a reference “(Department of Education and Science [DES] 1989a)”.

Reviewer 3 Report

Comments and Suggestions for Authors

The present review comprehensively analyze the advances of the iGluRs in the brain. A few concerns about the current version:

1) Please pay attention to the rules of the abbreviations.

2) In line 102-103, please provide the references for the sentence "pyramidal neurons, inhibitory interneurons, astrocytes, and cerebrovascular endothelial cells."

3) For the tables, three-line format is perfect for the publications.

4) For the legend in Fig 4, please delete the phrase "not shown". If it is not available in the picture, the authors should not to describe in the legend.

5) In line 837, the revealed is repeated.

6) In line 580, please check the sentence.

7) please avoid to cit the reference in the conclusion.
